# Membranous Nephropathy following Full-Dose of Inactivated SARS-CoV-2 Virus Vaccination: A Case Report and Literature Review

**DOI:** 10.3390/vaccines11010080

**Published:** 2022-12-29

**Authors:** Theerachai Thammathiwat, Laor Chompuk, Suchin Worawichawong, Vijitr Boonpucknavig, Supinda Sirilak, Sutatip Pongcharoen, Watchara Pichitsiri, Talerngsak Kanjanabuch

**Affiliations:** 1Division of Nephrology, Department of Medicine, Faculty of Medicine, Naresuan University, Phitsanulok 65000, Thailand; 2Department of Pathology, Faculty of Medicine, Naresuan University, Phitsanulok 65000, Thailand; 3Department of Pathology, Faculty of Medicine, Ramathibodi Hospital, Mahidol University, Bangkok 10400, Thailand; 4NHealth Company, Bangkok Hospital Group, Bangkok 10310, Thailand; 5Division of Immunology, Department of Medicine, Faculty of Medicine, Naresuan University, Phitsanulok 65000, Thailand; 6Division of Nephrology, Department of Medicine, Faculty of Medicine, Chulalongkorn University, Bangkok 10330, Thailand; 7Center of Excellence in Kidney Metabolic Disorders, Faculty of Medicine, Chulalongkorn University, Bangkok 10330, Thailand

**Keywords:** acute kidney injury, inactivated SARS-CoV-2 virus vaccination, membranous nephropathy, nephrotic syndrome

## Abstract

Vaccination against the SARS-CoV-2 virus (COVID-19) has proven to be the most effective measure to prevent the spread and reduce infection severity. A case report of de novo membranous nephropathy (MN) following immunization with inactivated virus vaccine (CoronaVac^®^, Sinovac Biotech) is presented here. A 53-year-old man presented with a sudden onset of leg edema a week after receiving an inactivated virus vaccine and a relapse of nephrotic syndrome (NS) with acute kidney injury (AKI) after a booster dose. Screening for serum anti-phospholipase A2 receptor antibody and secondary causes of MN were negative. Kidney biopsy revealed an early MN pattern with focal spike formation, whilst numerous subepithelial electron-dense deposits and a few small deposits in the mesangial area were observed through electron microscopy. A short course of steroids and oral cyclophosphamide was prescribed, resulting in the complete remission of NS and AKI. MN following SARS-CoV-2 vaccination should call for medical importance. Awareness of the association between vaccination and MN should be kept in mind to avoid unnecessary treatment with long-term immunosuppressive agents.

## 1. Background

Since the outbreak of the SARS-CoV-2 pandemic, the need for the SARS-CoV-2 vaccine has increased to alleviate the number of infections and reduce the severity of the disease [1]. Many studies have demonstrated the efficacy and safety of SARS-CoV-2 vaccines [2]. Four types of COVID-19 vaccines are being used worldwide including messenger RNA (mRNA), viral vector, protein subunit, and whole virus vaccines (known as inactivated virus vaccines). The inactivated virus vaccine elicits weaker immunogenicity and lowers clinical protection than the mRNA-based vaccine [3,4,5,6]. The inactivated viral vaccine (CoronaVac^®^, Sinovac Biotech) is produced by beta-propiolactone-inactivation of the CN2 strain of SARS-CoV-2 isolated from the patient’s bronchoalveolar lavage, which is closely linked to the 2019-nCoV-BetaCoV Wuhan/WIV04/2019 [1]. Results of phase 3 trials have shown that the inactivated viral vaccine has a high efficacy rate of 51–84% and a good safety profile [7,8,9]. Glomerular diseases following SARS-CoV-2 vaccines have been periodically reported including minimal change disease (MCD) and IgA nephropathy [10,11]. However, only a few cases of de novo or relapsed MN following inactivated viral vaccination have been documented [12,13]. A de novo MN with possible relapse stimulated by the second vaccination is reported here.

## 2. Case Presentation

A 53-year-old male patient presented with intermittent lower extremity edema and foamy urine for 2 weeks, which was spontaneously resolved. He received his first dose of the inactivated SARS-CoV-2 (CoronaVac^®^, Sinovac Biotech) vaccine a week before the onset of symptoms. He then had his second immunization after 4 weeks from the first dose. He experienced a sudden onset of leg and scrotal edema and puffy eyelids the next day after completing his primary vaccination series. He had abdominal discomfort and gained 5 kg of weight in one week. He denied any symptoms of gross hematuria, headache, or oliguria. Prior to admission, the SARS-CoV-2 virus was not detected by RT-PCR from the patient’s nasopharyngeal swab sample.

On physical examination, the vital signs were as follows: blood pressure 150/90 mmHg and heart rate 68 beats/min. His body mass index was 28.3 kg/m^2^. He had puffy eyelids without paleness and jaundice. His abdomen was distended with positive shifting dullness. Bilateral leg pitting edema and scrotal edema were also observed. His laboratory tests yielded serum creatinine 1.5 mg/dL, serum urea nitrogen 29 mg/dL, albumin 2.3 g/dL, cholesterol 507 mg/dL, and triglyceride 255 mg/dL. His urinary protein and erythrocytes levels were 3+ and 2+, respectively, and urine sediment depicted 3–5 per high-power field of red blood cells. His urine protein to creatinine ratio was 13.4 g protein per gram of creatinine. Tests for treponemal, HBsAg, anti-HCV, anti-HIV, and antinuclear antibodies were negative. Complement components C3 and C4 were within the normal values. Ultrasonography of the entire abdomen revealed that both kidneys were normal in shape and echogenicity.

He was clinically diagnosed with an acute onset of nephrotic syndrome (NS) associated with acute kidney injury (AKI). A kidney biopsy was performed, followed by a prescription of daily oral prednisolone at a dosage of 70 mg per day. The kidney biopsy finding showed 20 glomeruli with a normal glomerular basement membrane thickness. No glomerular proliferation and sclerosis were observed. Focal spike formation and scant subepithelial fuchsinophilic granules were detected in the Jones silver and Masson Trichrome stains, respectively, indicating an early stage of membranous pattern on light microscopy (LM). Diffuse interstitial edema was evidenced without accompanying tubular injury and interstitial inflammation. Immunofluorescence staining of 10 glomeruli showed diffuse granular deposition of the IgG (3+), C3 (3+), Kappa, and Lambda light chains (2+) along the capillary wall. MN was diagnosed (Figure 1).

Since the patient was classified as very high risk according to the KDIGO 2021 Clinical Practice Guideline for the Management of Glomerular Diseases (accompanying AKI not otherwise explained) [14], 3-day intravenous methylprednisolone (1 gm/day) was administered, followed by a 1.5-month course of oral prednisolone (0.5 mg/kg/day) and cyclophosphamide (2 mg/kg/day). In week 15, the electron microscopy (EM) report was issued denoting numerous subepithelial electron-dense deposits (EDDs) and a few small deposits in the mesangial, intramembranous, and subendothelial areas, suggesting secondary MN.

The serum anti-phospholipase A2 receptor (PLA2R) antibody (Ab) was negative. Other chronic infections and age-related malignancies were excluded. The suspicion of SARS-CoV-2 vaccine-induced secondary MN was raised; therefore, the steroid and immunosuppressive agent were replaced with angiotensin-converting enzyme inhibitors (ACEi). Complete remissions of proteinuria and abnormal kidney function were achieved after 2 weeks of discontinuation of the immunosuppression without subsequent relapses (Figure 2). 

## 3. Discussion 

A case of de novo MN with a possible disease flare or worsening of the clinical expression stimulated by the second vaccination was presented here with a favorable outcome, albeit presenting with AKI. Secondary MN was suspected because of the negative serum PLA2R Ab and the presence of mesangial EDDs. Chronic infections and malignancies had been ruled out. The patient fully recovered after a short course of steroid and immunosuppressive therapy.

Several glomerular diseases have been reported following SARS-CoV-2 vaccination [10,11]. The most commonly reported glomerulonephritis (GN) is MCD [10,11,15]. In a recent systematic review of vaccine-induced kidney adverse reactions from 90 case report articles, 134 cases of de novo GN were identified including MCD (52/134, 39%), IgA nephropathy (48/134, 36%), ANCA-associated GN (16/134, 12%), and MN (8/134, 6%) [16].

The mechanism of vaccine-induced MN remains inconclusive [17]. There is a case report of de novo MN and a case series of MN flares/relapses in patients with autoimmune disease following influenza H1N1 vaccinations [17,18]. An immunological response to the vaccination is postulated as the pathogenesis of the MN [17]. MN has been reported in association with all categories of the SARS-CoV-2 vaccine [10,11,18]. However, the mRNA-based vaccine has the highest number of case reports with vaccine-associated GN compared to the other vaccines, which is consistent with the finding of the highest immunogenicity of the vaccine [19,20]. In the systematic review of vaccine-induced kidney adverse reactions, mRNA vaccines contributed to 84% of the vaccine-associated GN cases, followed by viral vector vaccines (13%) and whole virus vaccines (3%) [16].

Recent case reports of de novo MN following vaccination with BNT162b2 (Pfizer-BioNTech) and exacerbation of MN after mRNA-1273 (Moderna) appear to support that vaccination with SARS-CoV-2 triggers MN [21]. Although the mechanism underlying the association has not yet been well-understood, two possible mechanisms have been proposed to explain how the vaccine may contribute to MN including vaccine-triggering genetically prone primary MN and vaccine-induced cross-immune response (molecular mimicry) [17,22,23]. In addition, numerous autoantibody-driven temporal pathogenesis in SARS-CoV-2 vaccine-associated MN have been postulated including autoantibodies to PLA2R [10,11,12,24], neural epithelial growth factor like-1 (NELL-1) [10], and intracellular proteins exostosin 1 (EXT1) [11]. However, Caza et al. presented cases with negatives of all of these antibodies [11]. As in the presented case, anti-PLA2R Ab was negative.

Table 1 demonstrates the pooled clinical spectra of SARS-CoV-2 vaccine-associated MN from the published literature. MN can occur after the first or boosted vaccination in both native [10,11,12,13,21,22,24,25,26,27] and kidney transplant recipients [28]. Clinical manifestation of MN following SARS-CoV-2 virus vaccination is usually full-blown NS with abrupt onset varying from 1 day to 4 weeks (Table 1).

As in this case, we reported acute onset of 1 week to develop symptoms following the first dose and a few days following the second dose of inactivated SARS-CoV-2 vaccines. The second episode might have resulted from a disease flare stimulated by the second vaccination. The early onset of NS has been mentioned in the passive Hayman nephritis, a classic model of human MN [29]. The MN is induced by a single injection of heterologous antisera to the rat renal tubular antigen extract in susceptible rat strains [23]. Massive proteinuria occurs in almost all animals within 5 days, followed by low-grade proteinuria lasting 60–150 days [23]. The onset of massive proteinuria is diminished by 2–3 days if the Ab is boosted [23]. Zhao et al. also demonstrated the occurrence of de novo MN immediately following immunization with the inactivated SARS-CoV-2 vaccine, and the NS entered partial remission after receiving angiotensin II receptor blocker treatment [13]. In the presented case, the abrupt onset of the NS was consistent with the early stage of membranous lesions through LM (focal spike formation) and EM (no membrane reaction).

The theory of vaccination triggering MN flare is supported by a retrospective study of 245 patients with biopsy-proven MN from a single center. The relapse rate of MN occurred at 5% during the SARS-CoV-2 pandemic era compared to 2% prior to the era [21]. However, the causal relationship of these findings needs further examination [11,30].

Treatment of secondary MN following SARS-CoV-2 vaccination is controversial. Based on the evidence of primary MN, conservative treatment and immunosuppressive medication are therapeutic options. Rituximab, obinutuzumab, mycophenolate mofetil, and oral cyclophosphamide have been used to reach partial response outcomes [10,19,21,24,25,26,27]. The patient achieved complete remission without subsequent second relapse after the short course of immunosuppressive agents, suggesting a causal relationship between the vaccination and secondary MN. Close monitoring of GN relapse after a booster dose of SARS-CoV-2 vaccination is warranted since the same type of vaccine might exacerbate the immunologic response [13,25]. 

In conclusion, secondary MN following SARS-CoV-2 vaccination should call for medical importance. Not only is it spontaneous remission or remission despite a short course of immunosuppressant, but also a high index of suspicion of the association might avoid unnecessary treatment with long-term steroid and immunosuppressive agents. Therefore, further investigation into the pathogenesis is warranted.

**Table 1 vaccines-11-00080-t001:** Clinical spectra and outcomes of membranous nephropathy following SARS-CoV-2 vaccination.

No.	Study	Age	G	Vaccine Type	Dose	Onset	Serum Albumin (g/L)	Serum Creatinine (mg/dL)	Hematuria (/HPF)	Proteinuria (g/g Creatinine)	De Novo/ Relapse GN	Type of MN	Treatment	Outcome
1.	This case	53	M	Sinovac	2nd	1 d	2.3	1.5	3–5	13.4	De novo	Neg only PLA2R	GC, CY 3 mo.	Response
2.	Aydin [12], 2021	66	F	Sinovac	1st	2 wk	2.6	2.78	N/A	9.42	Relapse	PLA2R	N/A	N/A
3.	Caza [11], 2021	54	M	Moderna	2nd	1 d	3.4	1.3	Pos	3+	De novo	1 PLA2R, 1 EXT, 1 neg PLA2R/ EXT	GC Rituximab	No response
4.		68	M	J + J	1st	<4 wk	3.2	3.3	Neg	0.6	De novo		Conservative	Partial response
5.		47	M	Moderna	2nd	6 d	2.3	0.7	Pos	2.7	De novo		None	Partial response
6.	Klomjit [10], 2021	50	F	Pfizer	2nd	4 wk	3.5	0.7	3–10	6.5	De novo	NELL-1	Conservative	Response
7.		39	M	Pfizer	2nd	1 wk	2	1.13	3–10	8.7	Relapse	PLA2R	TAC	Response
8.		70	M	Moderna	2nd	4 wk	2.7	2.1	<3	16.6	Relapse	PLA2R	Obinutuzumab	N/A
9.	Gueguen [25], 2021	76	M	Pfizer Moderna	1st 2nd	4 d N/A	1.6 2.2	0.86 1.15	Pos N/A	6.5 3.8	De novo Relapse	PLA2R N/A	Conservative/Rituximab	Partial response
10.	Da [31], 2021	70	M	Pfizer	1st	1 wk	1.7	1.29	N/A	4.4	De novo	THSD7A	Conservative	No response
11.	Liang [32], 2021	62	F	Moderna	2nd	1 mo	N/A	1.6	N/A	11.2	Relapse	PLA2R	Conservative/Rituximab	N/A
12.	Chavarot [28], 2022	66	M	Pfizer	2nd	8 wk	N/A	1.36	N/A	Neg	De novo post-KT	PLA2R	Conservative	N/A
13.	Psyllaki [24], 2022	68	M	Pfizer	1st	7 d	2.9	GFR 70 mL/min/1.73 m^2^	N/A	19	De novo	PLA2R	Rituximab	Partial response
14.	Fenoglio [22], 2022	82	F	Pfizer	2nd	88 d	N/A	N/A	N/A	NS	De novo	Neg PLA2R, THSD7A	GC	N/A
15.		67	F	Pfizer	2nd	89 d	N/A	N/A	N/A	NS	De novo	Neg PLA2R, THSD7A	Rituximab	N/A
16.		82	M	Pfizer	2nd	29 d	N/A	N/A	N/A	NS	De novo	PLA2R	Rituximab	N/A
17.	Rashid [27], 2022	56	M	Moderna	1st	4 wk	2.2	13.96	Blood 2+	12.2	De novo	PLA2R	Hemodialysis Rituximab	Response
18.	Visch [21], 2022	80	M	Pfizer	2nd	4 wk	2.6	1.32	N/A	5	Relapse	PLA2R	Rituximab	No response
19.		60	M	Pfizer	2nd	6 wk	1.7	1.92	N/A	5	Relapse	PLA2R	Rituximab/CY/ GC	Response
20.		77	F	Pfizer	1st	4 wk	2.2	0.7	N/A	12.5	Relapse	PLA2R	Tacrolimus	Response
21.		78	M	Pfizer	2nd	1 wk	3.4	1.87	N/A	4.9	Relapse	N/A	GC	Response
22.		48	M	Pfizer	2nd	3 wk	3.1	1.41	N/A	1.7	Relapse	PLA2R	Conservative	No response
23.		56	M	Pfizer	2nd	2 wk	3.2	1.47	N/A	3.4	Relapse	PLA2R	Conservative	No response
24.		84	M	Pfizer	2nd	10 wk	3.3	1.55	N/A	3	Relapse	PLA2R	Tacrolimus/GC→ Rituximab	Response
25.		39	M	Pfizer	2nd	4 wk	1.8	1.38	N/A	3.7	De novo Worsening	PLA2R	Rituximab/CY/ GC	Response
26.		75	M	Pfizer	2nd	2 wk	2.1	0.88	N/A	8	De novo Worsening	PLA2R	Rituximab/CY/ GC	N/A
27.		48	M	Pfizer	1st	2 wk	2.5	1.26	N/A	2.22	De novo Worsening	PLA2R	Rituximab/CY/ GC	Response
28.		58	M	Pfizer	2nd	3 wk	2.4	1.02	N/A	8	De novo Worsening	PLA2R	Tacrolimus	Response
29.	Zhao [13], 2022	57	W	Sinovac	1st 2nd	1 d 1 d	N/A 2.85	N/A 0.42	N/A 1+	N/A 1.6	De novo Relapse	PLA2R	Conservative	Partial response
30.	Paxton [26], 2022	22	M	Pfizer	2nd	4 wk	8	0.72	N/A	7	De novo	PLA2R	Rituximab	Partial response
31.	Saigal [33], 2022	32	M	Astra Zeneca	N/A	14 d	N/A	N/A	N/A	N/A	De novo	N/A	GC/CY	Partial response
32.		47	M	Astra Zeneca	N/A	11 d	N/A	N/A	N/A	N/A	De novo	N/A	Conservative	Response
32.	Pitre [34], 2022	65	F	J + J	1st	5 mo	N/A	1.7	N/A	1.7	De novo	PLA2R	GC	Partial response
33.	Ma [19], 2022	42	F	Astra Zeneca	1st	2 wk	1.6	0.8	N/A	16	N/A	N/A	GC/MMF	Response

Abbreviations: G, gender; M, male; F, female; GC, glucocorticoids; CY, cyclophosphamide; MMF, mycophenolate mofetil; PLA2R, M-type phospholipase A2 receptor antibody; NELL-1, neural epithelial growth factor like-1; EXT1, intracellular proteins exostosin; THSD7A, thrombospondin type 1 domain containing 7A; NS, Nephrotic syndrome; J + J, Johnson & Johnson SARS-CoV-2 vaccine; N/A, not applicable.

## Figures and Tables

**Figure 1 vaccines-11-00080-f001:**
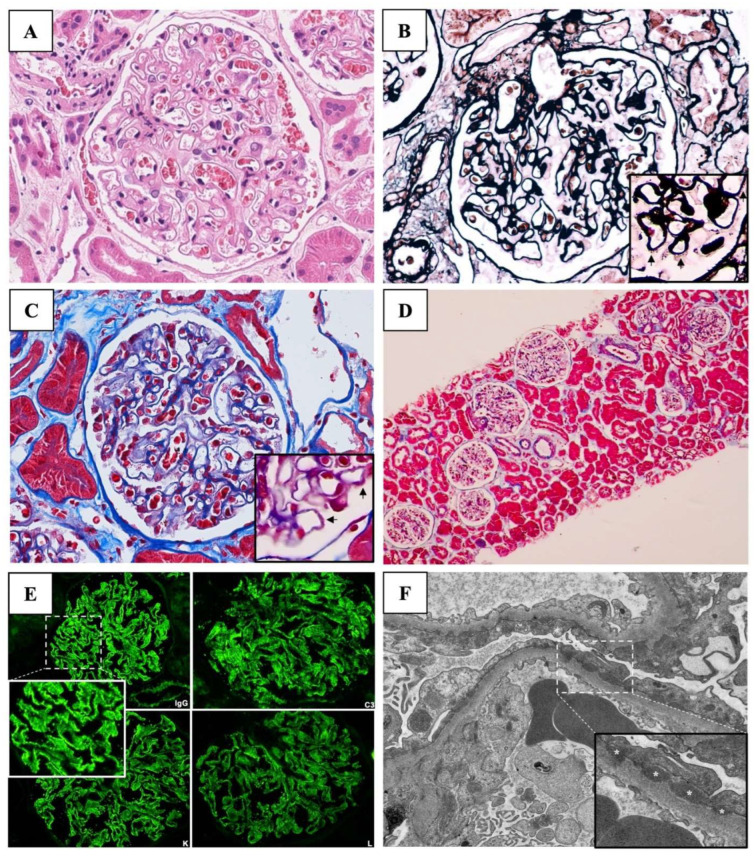
Kidney biopsy findings. (**A**) Light microscopy of the glomerulus with Hematoxylin and Eosin (original magnification, ×200) revealed normal glomeruli. (**B**) Jones silver stain (original magnification, ×200) showed focal spike formation on the glomerular basement membranes (arrows). (**C**) Masson trichrome stain (original magnification, ×200) depicts the subepithelial fuchsinophilic granules (arrows). (**D**) Masson trichrome stain (original magnification, ×4) revealed diffuse interstitial edema without accompanying tubular injury and interstitial inflammation. (**E**) Direct immunofluorescent studies demonstrate diffuse granular deposition with strongly positive (3+) stains for IgG, C3, kappa, and lambda with negative stains of IgA, IgM, and C1q. (**F**) Electron microscopy (original magnification, ×5000) revealed numerous subepithelial electron-dense deposits (asterisks) and a few small deposits in mesangial, intramembranous, and subendothelial area with podocyte foot process effacement (magnification, ×10,000). * denotes electron-dense deposits under electron microscopy.

**Figure 2 vaccines-11-00080-f002:**
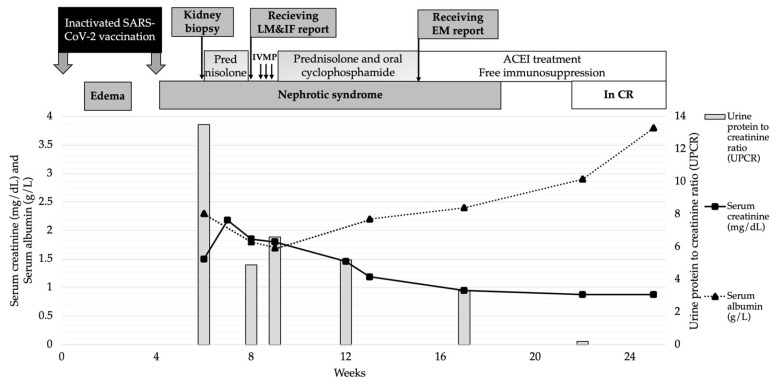
Patient’s clinical course and treatment. Abbreviations: IVMP, intravenous methyl prednisolone; MKD, mg/kg/day; CR, complete remission.

## Data Availability

Not applicable.

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
