# Peer review of "Membranous Nephropathy following Full-Dose of Inactivated SARS-CoV-2 Virus Vaccination: A Case Report and Literature Review"

_vaccines, 2022, doi:10.3390/vaccines11010080_

Round 1
Reviewer 1 Report
The authors describe a patient who developed edema after one week of the first administration of an inactivated SARS-CoV-2 vaccine and a nephrotic syndrome (NS) one day after the administration of the second dose one month later. The renal biopsy showed a membranous glomerulonephritis (MN) which seemed not secondary to other common causes. A mild renal insufficiency was detected. The patient received a therapy with immunosuppression and a complete recovery is reported after five months.
Several cases of nephrotic syndrome detected after anti Covid vaccinations have been reported. Most cases received a mRNA vaccine and had pathological lesions comparable with the diagnosis of minimal change disease; the NS compared within one week from the vaccine administration and the hypothesis is that vaccines induce a cell-mediated immune response through recognition by CD8+ T cells, while CD4+ cells promote the production of antigen-specific antibodies by B cells. In the cases of MN it is hard to believe that the renal lesions may be produced in a such short time and the NS may be due to a relapse stimulated by the vaccine.
In the case described in the article I would not exclude a relapse or worsening of the clinical expression of a MN. The kidney biopsy findings do not show the interstitium and the glomerular lesions are mild and should be better illustrated; no spikes are visible and the granular deposition on IF and the deposit on EM are hard to see. A reduction of GFR, although mild, is not compatible with these findings.
The authors should justify the immunosuppressive therapy since it is not supported by the international guidelines and not recommended if the MN was believed secondary to the vaccine.
Author Response
Reviewer #1
The authors describe a patient who developed edema after one week of the first administration of an inactivated SARS-CoV-2 vaccine and a nephrotic syndrome (NS) one day after the administration of the second dose one month later. The renal biopsy showed a membranous glomerulonephritis (MN) which seemed not secondary to other common causes. A mild renal insufficiency was detected. The patient received a therapy with immunosuppression and a complete recovery is reported after five months.
- Several cases of nephrotic syndrome detected after anti Covid vaccinations have been reported. Most cases received a mRNA vaccine and had pathological lesions comparable with the diagnosis of minimal change disease; the NS compared within one week from the vaccine administration and the hypothesis is that vaccines induce a cell-mediated immune response through recognition by CD8+ T cells, while CD4+ cells promote the production of antigen-specific antibodies by B cells. In the cases of MN it is hard to believe that the renal lesions may be produced in a such short time and the NS may be due to a relapse stimulated by the vaccine. In the case described in the article, I would not exclude a relapse or worsening of the clinical expression of a MN.
Response: We agree that the second episode or the exacerbation of NS might result from a relapse or worsening of the clinical expression stimulated by the second vaccination. The discussion section has been revised accordingly as below.
"The early onset of NS has been mentioned in the passive Hayman nephritis, a classic model of human MN [29]. The MN is induced by a single injection of heterologous antisera to rat renal tubular antigen extract in susceptible rat strains [23]. Massive proteinuria occurs in almost all animals within 5 days, followed by low-grade proteinuria lasting 60-150 days [23]. The onset of massive proteinuria is diminished by 2-3 days if the Ab is boosted [23]."
- The kidney biopsy findings do not show the interstitium and the glomerular lesions are mild and should be better illustrated; no spikes are visible and the granular deposition on IF and the deposit on EM are hard to see. A reduction of GFR, although mild, is not compatible with these findings.
Response: The magnification views of silver staining, IF, and EM have been added to the Figures accordingly. The case presentation section has been revised accordingly as below. Diffuse interstitial edema was evidenced in the kidney pathology without accompanying tubular injury and interstitial inflammation. Thus, the reduction of GFR might be explained by nephrosarca or hypovolemia due to severe hypoalbuminemia.
"Focal spike formation and scant subepithelial fuchsinophilic granules were detected in Jones silver and Masson Trichrome stains, respectively, indicating an early stage of membranous pattern on light microscopy (LM)."
- The authors should justify the immunosuppressive therapy since it is not supported by the international guidelines and not recommended if the MN was believed secondary to the vaccine.
Response: The case presentation section has been revised accordingly as below.
"Since the patient was classified as very high risk according to the KDIGO 2021 Clinical Practice Guideline for the Management of Glomerular Diseases (accompanying with AKI not otherwise explained) [14], 3-day intravenous methylprednisolone (1 gm/day) were administered, followed by a 1.5-month course of oral prednisolone (0.5 mg/kg/day) and cyclophosphamide (2 mg/kg/day)."
Reviewer 2 Report
Comments for Authors
Dear Authors
Thank you for choosing “vaccines for publishing your work.
The manuscript is well written and structured and can be published after some revisions
Comment 1. In the introduction you should cite relevant studies to support the statement: “Many studies have demonstrated the efficacy and safety of SARS-CoV-2 virus vaccines”.
Comment 2. The order of citations as they appear in the document is not correct. For example the following statement of the introduction: “Results of phase 3 trials have shown that inactivated SARS-CoV-2 vaccine has a high efficacy rate of 50.7 –83.5% and a good safety profile2-4.” Citations 2 and 3 refer to glomerulonephritis after covid vaccination and citations 5-6 refer to efficacy of the new vaccine and they are not cited in this sentence. The next sentence in the manuscript: “The glomerular 50 diseases following SARS-CoV-2 vaccines were reported periodically, including minimal change disease and IgA nephropathy 5, 6” has the wrong citations.
Comment 3. You should dedicate at least a paragraph in the introduction of the manuscript where you should elaborate on the different types of covid vaccines (mRNA vs. Vector based vs. inactivated). You should refer to the immunogenicity of each vaccine type and how it is compared to each other based on the studies that have already been published which compared the immune response between the different vaccines including the CoronaVac vaccine.
Comment 4. In the introduction you mention that has been one more case of membranous after the CoronVac vaccine. Firstly, you should clarify that the case report you cite was relapse of membranous and not new onset as in the case you present in this manuscript. Second, this is not the only case report already published: In June 2022 there was another case report was published on a patient with new onset membranous after the CoronaVac vaccination (Zhao Y, Zhang L, Wang G, Guan J, Pai P. The Development of De novo Acute Tubulointerstitial Nephritis and Membranous Nephropathy Following Inactivated COVID-19 Vaccine: Causal or Casual?. Clin Case Rep Int. 2022; 6: 1344.). A more comprehensive literature search should be done to cite similar case reports.
Comment 5.
In the discussion you should discuss the degree of immunogenicity of each type of covid vaccine (mRNA vs. vector based vs. inactivated virus) and how it may be related to the incidence of de-novo glomerulonephritis after vaccination considering of course the percentage of population that has received each vaccine according to the available data.
Comment 6
In the discussion, you should also briefly mention the case reports of other de-novo glomerular diseases after the CoronaVac vaccination
Author Response
Reviewer 2
Thank you for choosing “vaccines” for publishing your work. The manuscript is well written and structured and can be published after some revisions.
- In the introduction, you should cite relevant studies to support the statement: "Many studies have demonstrated the efficacy and safety of SARS-CoV-2 virus vaccines".
Response: The appropriate citation has been added to the sentence.
- The order of citations as they appear in the document is not correct. For example, the following statement of the introduction: “Results of phase 3 trials have shown that inactivated SARS-CoV-2 vaccine has a high efficacy rate of 50.7 –83.5% and a good safety profile2-4.” Citations 2 and 3 refer to glomerulonephritis after covid vaccination and citations 5-6 refer to efficacy of the new vaccine and they are not cited in this sentence. The next sentence in the manuscript: “The glomerular 50 diseases following SARS-CoV-2 vaccines were reported periodically, including minimal change disease and IgA nephropathy 5, 6” has the wrong citations.
Response: Please accept our apology for the mistakes. The references have been edited accordingly.
- You should dedicate at least a paragraph in the introduction of the manuscript where you should elaborate on the different types of covid vaccines (mRNA vs. Vector based vs. inactivated). You should refer to the immunogenicity of each vaccine type and how it is compared to each other based on the studies that have already been published which compared the immune response between the different vaccines including the CoronaVac vaccine.
Response: Thank you for your valuable comment. The introduction section has been revised accordingly as below.
“Four types of COVID-19 vaccines are being used worldwide, including messenger RNA (mRNA), viral vector, protein subunit, and whole virus vaccines (known as inactivated virus vaccines). Inactivated virus vaccine elicits weaker immunogenicity and lowers clinical protection than the mRNA-based vaccine [3-6]."
- In the introduction you mention that has been one more case of membranous after the CoronVac vaccine. Firstly, you should clarify that the case report you cite was relapse of membranous and not new onset as in the case you present in this manuscript. Second, this is not the only case report already published: In June 2022 there was another case report was published on a patient with new onset membranous after the CoronaVac vaccination (Zhao Y, Zhang L, Wang G, Guan J, Pai P. The Development of De novo Acute Tubulointerstitial Nephritis and Membranous Nephropathy Following Inactivated COVID-19 Vaccine: Causal or Casual?. Clin Case Rep Int. 2022; 6: 1344.). A more comprehensive literature search should be done to cite similar case reports.
Response: We add your suggestion in the introduction section accordingly.
“However, only a few cases of de-novo or relapsed MN following inactivated viral vaccination have been documented [12,13].”
- In the discussion you should discuss the degree of immunogenicity of each type of covid vaccine (mRNA vs. vector based vs. inactivated virus) and how it may be related to the incidence of de-novo glomerulonephritis after vaccination considering of course the percentage of population that has received each vaccine according to the available data.
Response: The discussion section has been revised accordingly as below.
“However, the mRNA-based vaccine has the highest number of case reports with vaccine-associated GN compared to the other vaccines, which is consistent with the finding of the highest immunogenicity of the vaccine [19,20]. In the systematic review of vaccine-induced kidney adverse reactions, mRNA vaccines contribute to 84% of the vaccine-associated GN cases, followed by viral vector vaccines (13%) and whole virus vaccines (3%) [16].”
- In the discussion, you should also briefly mention the case reports of other de-novo glomerular diseases after the CoronaVac vaccination
Response: Thank you for your recommendation. We add this suggestion to the discussion section accordingly.
“Zhao et al. have also demonstrated the occurrence of de-novo MN immediately following immunization with the inactivated SARS-CoV-2 vaccine, and the NS entered partial remission after receiving angiotensin II receptor blocker treatment [13].”
Round 2
Reviewer 1 Report
Thank you for you revision